# An Analytical Geometry Optimization Model for Current-Mode Cross-Like Hall Plates

**DOI:** 10.3390/s19112490

**Published:** 2019-05-31

**Authors:** Yue Xu, Xingxing Hu, Lei Jiang

**Affiliations:** 1College of electronic and optical engineering & College of microelectronics, Nanjing University of Posts and Telecommunications, Nanjing 210023, China; janet_hxx@163.com (X.H.); 1218023011@njupt.edu.cn (L.J.); 2National and Local Joint Engineering Laboratory of RF Integration and Micro-Assembly Technology, Nanjing 210023, China

**Keywords:** current-mode Hall device, length-to-width ratio (*L*/*W*), current sensitivity, signal-to-noise ratio (SNR)

## Abstract

This paper presents a new analytical geometry optimization model to depict the optimal current sensitivity and signal-to-noise ratio (SNR) for the current-mode Hall devices. The conformal mapping calculation is performed to study the influence of device geometry on the current sensitivity and SNR of the current-mode cross-like Hall plates. The analytical model indicates that a current-mode cross-like Hall plate can achieve optimal current sensitivity and SNR in the device length-to-width ratio (*L*/*W*) range of 0.4–0.5 when the thermal noise is taken into account. Three-dimensional (3D) technology computer aided design (TCAD) simulation validates the accuracy of the analytical model. The proposed analytical model provides a geometry design rule to achieve optimal sensitivity and SNR at the same time for the current-mode cross-like Hall plates.

## 1. Introduction

Silicon-based Hall sensors have been widely applied in many fields such as control systems, consumer electronics, and automobiles due to its low cost, high integration and high reliability [1,2,3]. Generally, the Hall devices are biased by a constant voltage source or a current source and sensed by the Hall voltage signal. However, this traditional output-voltage working mode suffers low magnetic sensitivity and high noise [4]. A Hall device under current mode that can achieve a twice-higher magnetic sensitivity than that working in voltage mode has been introduced to obtain a better signal-to-noise ratio (SNR) [5,6,7]. In order to further enhance the performance of the current-mode Hall devices, it is essential to study the influence of device geometric parameters, including the length and width of the active area and the bias electrodes on the sensitivity and noise of the Hall devices. 

Recently, a great deal of research work was carried out to optimize the geometry of the voltage-mode Hall device for better performance in magnetic field detection. Lyu et al. analyzed the magnetic sensitivity and offset characteristics dependence of different covering layers for cross-like Hall plates [8]. Crescentini et al. focused on the effects of the contact positions of the square Hall plate on the sensitivity, power consumption, and bandwidth [9]. Zhang et al. proposed a geometry optimization method to improve the sensitivity of the planar Hall devices by means of the conformal mapping technique [10]. However, although all of them have taken into account the effect of different structures on sensitivity, the device geometry optimization for the maximum SNR was not studied. It is worth noting that the signal processing circuits in Hall sensors are highly limited by offset and noise but not by small Hall signals, therefore the maximum SNR should be achieved through device geometry optimization design. To this end, Zhang et al. investigated the low-frequency 1/f noise regarding the different device dimensions and doping levels for the cross-like Hall plates. It is found that a larger active area with an optimal cross length-to-width ratio (*L*/*W*) is beneficial for a higher SNR [11]. In fact, the 1/f noise can be chopped out by the spinning current scheme in the switched Hall sensors, therefore, the thermal noise of a cross-like Hall plate should be considered first. Aiming at the thermal noise of Hall devices, Ausserlechner derived an expression for the optimal SNR, which allows for an analytical optimization of device geometry. It is suggested that the Hall plates with maximum SNR should have symmetrical characteristics with equal input and output resistances [12]. However, the above-mentioned sensitivity and SNR optimization methods are all proposed for the voltage-mode Hall devices, and the geometry optimization strategy for current-mode devices has not been reported yet. 

In this paper, an analytical geometry optimization model is established to predict the optimal performance for the current-mode cross-like Hall plates by using the conformal mapping method. The geometry optimization of the cross-like Hall plate operating at current mode is proposed to attain the optimal sensitivity and SNR at the same time. The manuscript is organized as follows: Section 2 introduces the basic theory of the current-mode Hall plate. In Section 3, the geometry optimization model is presented based on the conformal mapping technique. In Section 4, the proposed model was verified by three-dimensional (3D) technology computer aided design (TCAD) simulation. Section 5 summarizes all of the work.

## 2. Basic Theory of Current-Mode Hall Plate

In the current-mode operation, the two biasing terminals of Hall devices are input by a current source and a current sink with the same value, respectively, which maintains the equal potential at the two sensing terminals. As a result, a differential Hall current appears on the two sensing terminals when an external magnetic field is applied. The applied magnetic field changes the current flow in the Hall devices, leading to a variation of the sheet resistances by ΔR_I_. Similarly, in the voltage-mode Hall devices, the applied magnetic field alters the value of sheet resistances by ΔR_V_. A four-resistance Wheatstone bridge model is used to describe the DC electrical relationship between the biasing and sensing terminals, as shown in Figure 1. 

In the voltage-mode operation shown in Figure 1a, if the Hall plate is biased with a voltage source *V_bias_* under an external vertical magnetic field *B*, the output voltage at two sensing terminals can be calculated respectively by
(1)$$V_{o1}=\;\frac{V_{bias}}{2R}(R-\Delta R_{V})$$
(2)$$V_{o2}=\;\frac{V_{bias}}{2R}(R+\Delta R_{V})$$

Then, the differential Hall voltage is given by
(3)$$V_{H}=V_{o1}-V_{o2}=\;V_{bias}\frac{\Delta R_{V}}{R}$$

Hence, the voltage-mode sensitivity at magnetic flux density *B* = 1 T can be expressed by
(4)$$S_{V,V}=\;\frac{V_{H}}{B\cdot V_{bias}}=\frac{\Delta R_{V}}{R}\cdot T^{-1}$$

As for the current-mode operation illustrated in Figure 1b, a bias current source *I_bias_* is input into node C_1_, and a bias current sink *I_bias_* with the same value is drawn from node C_3_. Nodes C_2_ and C_4_ have the same ground-to-ground common-mode voltage. Accordingly, the differential output currents are respectively given by
(5)$$I_{o1}=\frac{\Delta R_{I}}{R}\cdot I_{bias}$$
(6)$$I_{o2}=-\frac{\Delta R_{I}}{R}\cdot I_{bias}$$

Then, the Hall current is equal to
(7)$$I_{H}=\;I_{o1}-I_{o2}=2I_{bias}\frac{\Delta R_{I}}{R}$$

If we suppose the same resistance change Δ*R_V_* = Δ*R_I_* for simple approximation, the sensitivity in the current mode is twice larger than that in the voltage mode. Finally, the current-mode sensitivity can be approximately expressed as
(8)$$S_{I,I}=\frac{I_{H}}{B\cdot I_{bias}}≅\;2S_{V,V}$$

On the other hand, the voltage-mode sensitivity is given by [13]
(9)$$S_{V,V}=\;\mu_{H}\frac{G}{N_{square}}$$

Here, *μ_H_* is the Hall mobility and *G* is the geometrical correction factor of the Hall plate and $N_{square}$ is the number of squares of sheet resistivity. The sheet resistivity, also called square resistance, is equal to $R_{square}=\frac{1}{q\mu Nt}$, with *μ* the carrier mobility and *N* the doping level and *t* the thickness of Hall plate.

Therefore, the current-mode sensitivity can be written by
(10)$$S_{I,I}=2\mu_{H}\frac{G}{N_{square}}$$

## 3. Geometry Optimization Model

### 3.1. Conformal Mapping Calculation of Geometrical Correction Factor

The conformal mapping method was used to derive the analytical expression of the geometrical correction factor *G*. Figure 2 illustrates the schematic of the calculation procedure of *G*. First, we mapped the cross-like device structure in the s-plane onto the unit disk in the z-plane. For a given device geometry with vertices S_1_, …, S_8_ the transformation between the s-plane and the z-plane is performed by [14]
(11)$$s=g(z)=C{\int_{0}^{z}{\left(1+\varepsilon^{4}\right)}^{1/2}\left[\varepsilon^{8}-2\varepsilon^{4}\cos(4\theta )+1\right]}^{-1/2}d\varepsilon$$
where the angle *θ* represents half of the unit disk contact, as shown in Figure 2. Parameter *C* is a function of *θ*, which can be calculated numerically. The boundary points Z_1_, …, Z_8_ of the unit disk correspond to the vertices S_1_, …, S_8_ of the cross-like shape. Furthermore, the relationship between the two geometries satisfies the following equation [15]
(12)$$\theta =2\sqrt{2}\exp\left[-\pi (\frac{1}{4}+\frac{L}{W})\right]$$

Secondly, the unit disk of the z-plane is mapped to the polygon area of the w-plane by Schwarz–Christoffel transformation (SCT):(13)$$W=\;f(Z)=C_{1}\int_{0}^{Z}\prod_{i=1}^{8}{\left(1-\frac{\xi }{Z_{i}}\right)}^{\frac{\alpha_{i}}{\pi }-1}\cdot (1-\frac{\xi }{M_{1}})\cdot (1-\frac{\xi }{M_{2}})d\xi +C_{2}$$
where the parameters *C*_1_ and *C*_2_ denote the expansion coefficient and transformation center, respectively. The boundary points *Z*_1_, …, *Z*_8_ of the unit disk correspond to the vertices *W*_1_, …, *W*_8_ of the rhombus. The parameters of *α*_1_π, …, *α*_8_π are the interior angles of the polygon. *M*_1_ and *M*_2_ represent the points at which the current flows in and out, respectively. If we label *W_k_* = *f*(*Z_k_*) and *M*_k_’ = *f*(*M_k_*), the edge length of the polygon *W_k_W_k_*_+1_ can be calculated by
(14)$$\left|W_{k}-W_{k+1}\right|=\left|f(Z_{k})-f(Z_{k+1})\right|$$
in which, *Z_k_* = *e*^i*φk*^, *α_k_* and *φ_k_* are summarized in Table 1.

Based on such conformal mapping, the Hall voltage can be calculated through changing the real device geometry into simple edge length ratios of a polygonal region [16]. In the polygonal region, the electric field $\vec{E}$ and current density $\vec{J}$ are uniform, and $\vec{E}$ is perpendicular to *W*_1_*W*_8_, but $\vec{J}$ is parallel to *W*_1_*W*_4_. In this case, the Hall voltage can be calculated by
(15)$$V_{H}=(\left|W_{7}W_{8}\right|-\left|W_{1}W_{2}\right|)\cos\beta$$

On the other hand, one finds [17]
(16)$$I_{bias}R_{□}=\left|W_{1}W_{4}\right|\cos^{2}\beta$$

Together with Equations (15) and (16), we can obtain
(17)$$\frac{V_{H}}{I_{bias}}=\frac{\left|W_{7}W_{8}\right|-\left|W_{1}W_{2}\right|}{\left|W_{1}W_{4}\right|}\frac{R_{square}}{\cos\beta }$$

As is well known, the Hall voltage is expressed by
(18)$$V_{H}=GR_{square}\mu_{H}BI_{bias}$$

Substituting Equation (18) into Equation (17) and using the equation *μ_H_B* = tan*β*, the geometrical correction factor can be written as follows:(19)$$G=\frac{\left|W_{7}W_{8}\right|-\left|W_{1}W_{2}\right|}{\left|W_{1}W_{4}\right|}\frac{1}{\sin\beta }$$

According to Equations (12), (13), (14) and (19), the geometrical correction factor *G* as a function of the *L*/*W* ratio at different Hall angles were calculated and the *G*-*L*/*W* curves are displayed in Figure 3. It is found that the geometrical correction factor is increased from 0 to 1 with the *L*/*W* increasing. We further notice that the geometrical correction factor becomes almost unrelated to small Hall angles, namely, low magnetic field. As shown in the inset of Figure 3, it is obvious that the differences between the *G* values can be negligible when the Hall angle *β* is changed between 0.1° and 2°. For a low doped silicon Hall device, *μ_H_* = 1000 cm^2^/V·s and a low magnetic field *B* = 0.4 T, the Hall angle is less than 2°. Consequently, an analytical expression of the geometrical correction factor independence of Hall angle can be extracted for a cross-like Hall plate operating at low magnetic field less than 0.4 T. Using MATLAB, we fit the model calculation curves under low magnetic field with an exponential function as follows:(20)$$G=1-1.044\exp(-3.142\frac{L}{W})$$

If 0 ≤ *L*/*W* ≤ 3.5, the relative deviation between the *G* model and the fitted Equation (20) is less than 0.1%.

### 3.2. Optimal Sensitivity

For a cross-like Hall plate, the number of squares of input resistance is approximately equal to [13]:(21)$$N_{square}≅\;2\frac{L}{W}+\frac{2}{3}$$

Substituting Equation (21) into Equation (10), we have
(22)$$S_{I,I}≅\;\frac{2G}{2\frac{L}{W}+\frac{2}{3}}\mu_{H}$$

For example, a typical 0.18 μm standard Complementary Metal Oxide Semiconductor (CMOS) technology, the doping concentration in the N-well active area is about 2 × 10^17^ cm^−3^. Under this doping level, the electron Hall mobility is nearly 700 cm^2^/V·s and the Hall scattering factor for electrons is about 1.1 [18]. In term of Equations (20) and (22), the current-mode sensitivity as a function of *L*/*W* ratio was calculated and the model calculated results are shown in Figure 4. It is seen that when the value of *L*/*W* ratio increases from 0 to 3.5, the current-mode sensitivity rises sharply and it reaches a maximum value of about 6.8%/T at *L*/*W* ≈ 0.37. After that, the current-mode sensitivity shows a relatively slow downward trend. Apparently, the current-mode sensitivity can always keep the optimal value as the *L*/*W* ratio is within 0.3–0.5.

### 3.3. Optimal Signal-to-Noise Ratio (SNR)

Besides the Hall sensing performance, the noise of a Hall device is also an important issue of concern. Severe noise will overwhelm the weak Hall signal, making the Hall device inoperable. The noise in Hall devices mainly includes low-frequency 1/*f* noise and thermal noise. It is worth noting that the offset and low-frequency 1/*f* noise of Hall devices can be effectively eliminated by the spinning current circuit. As the Hall device is chopping at high frequency, the thermal noise becomes the determinant of SNR. Therefore, we just need to take into account the thermal noise in the switched Hall sensors. The thermal noise voltage between two sensing terminals is characterized by
(23)$$V_{noise}=\;\sqrt{4k_{B}TR_{out}\Delta f}.$$
with *k_B_* the Boltzmann constant, *T* the absolute temperature, *R_out_* the output resistance, and Δ*f* the effective noise bandwidth. Therefore, the SNR of the current-mode cross-like Hall plate can be calculated using Equations (22) and (23):(24)$$SNR=\;\frac{I_{H}}{I_{noise}}=\frac{I_{bias}\cdot S_{I,I}\cdot B}{\frac{\sqrt{4k_{B}TR_{out}\Delta f}}{R_{out}}}=K\cdot \frac{G}{{\left(2\frac{L}{W}+\frac{2}{3}\right)}^{\frac{1}{2}}}.$$
where $R_{out}=\;(2\frac{L}{W}+\frac{2}{3})R_{square}$ and $K=\;2I_{bias}B\mu_{H}\sqrt{\frac{R_{square}}{4k_{B}T\Delta f}}$, which can be regarded as a constant.

Figure 5 shows the trend of the calculated SNR varying with *L*/*W* ratio. It is found that the current-mode SNR is proportional to the bias current. However, a large bias current can cause a large offset current, thus the value of the bias current should be limited in the practical applications. At a fixed bias current, the maximum value of SNR is achieved at *L*/*W* ≈ 0.6. Furthermore, an optimal SNR of the cross-like Hall plate can be acquired when the *L*/*W* ratio is in a relatively wide range from 0.4 to 0.8. 

When the influence of the cross shape of Hall plate on the current sensitivity and SNR are considered concurrently, it is suggested that the *L*/*W* ratio of 0.4–0.5 is a beneficial geometry parameter to achieve the optimal current-mode sensitivity and SNR at the same time. Using the proposed analytical model, the optimal performance of the current-mode Hall plate can be expected by geometry optimization.

## 4. TCAD Simulation Verification

To verify the accuracy of the analytical geometry optimization model, three-dimensional (3D) TCAD simulation was carried out using Silvaco Atlas tool. First of all, based on a standard 0.18 μm process, a 3D device simulation structure was established according to the process and geometric parameters such as the distribution of N-well doping concentration, N-well depth, contact size, N-well size, etc. Secondly, the device DC simulation was carried out where the two bias currents of the same size are applied to the input terminals, and the output Hall signal and noise are measured between the sensing terminals. Finally, the current sensitivity and signal-to-noise ratio (SNR) was calculated according to the simulation data. In the current-mode simulation, appropriate models, such as Shockley-Read-Hall recombination, low-field mobility, magnetic, etc., were used. Based on SMIC 0.18-μm standard CMOS technology, the Hall current and thermal noise variations with the cross shape of Hall plate were studied. The simulated 3D schematic of the cross-like Hall plate is illustrated in Figure 6a. A cross-shaped N-well active area is formed on the P-type silicon substrate. Four N+ contacts are located at the four cross regions with 90° symmetrical structure and each of them is extended to the boundary of the N-well to reduce the offset of the device. The simulated net dopant profile as a function of depth in the active layer is shown in Figure 6b. It can be seen that the doping level of N-well is about 2 × 10^17^ cm^−3^ and the depth of Nwell is about 3 μm, which well accords with the distribution of doping concentrations in the actual process.

Figure 7a shows the simulated current-mode sensitivity versus different *L*/*W* ratios at *I*_bias_ = 1 mA and *B* = 5 mT. TCAD simulation results show that the current-mode sensitivity increases and then decreases as the *L*/*W* ratio is increased. A maximum current-mode sensitivity of about 6.7%/T appears when the ratio of *L*/*W* is nearly 0.4 under the low magnetic field, which is in agreement with the model calculation results. Compared with the finite element modeling (FEM) simulation presented in the literature for cross-like Hall sensors [4], the obtained magnetic sensitivity from TCAD simulation shows higher accuracy. This is due to the fact that the key model parameters used in TCAD simulation, such as N-well conductivity, electrons concentration, or Hall coefficient, can be accurately calculated according to the distribution of impurity concentration and device material characteristics. We further find that when the *L*/*W* is in the range of 0.3–0.5, the cross-like Hall plate can obtain the optimal current-mode sensitivity. Additionally, it is shown that the current-mode sensitivity is about twice greater than the voltage-mode sensitivity (see Figure 7b), which validates the correctness of the model analysis.

The SNR characteristic of the current-mode cross-like Hall plate was analyzed using the simulated current-mode sensitivity and thermal noise data. Figure 8 illustrates the simulated SNR with respect to the different *L*/*W* ratios. It is found that the cross-like Hall plate achieves a maximum SNR when the *L*/*W* ratio is around 0.6, meanwhile, the SNR can keep an optimal value as the ratio of *L*/*W* is ranged from 0.5 to 1. The TCAD simulation results show the same tendency with the theoretical model calculation, proving the accuracy of the proposed analytical optimization model.

The current-mode sensitivity is highly affected by the geometrical correction factor, the Hall mobility and the square number of input resistance, but is less affected by the bias current. The Hall mobility depends on the concentration of impurities for a given semiconductor material, while the geometrical correction factor and the input resistance square number are directly related to the *L*/*W* ratio, therefore, the current-mode sensitivity is dominated by the geometry parameter of *L*/*W* for cross-like Hall plate. Both the model calculation results and the TCAD simulation data indicate that the optimal current-mode sensitivity can be acquired when the *L*/*W* of a cross-like Hall plate is in the range of 0.3–0.5. On the other hand, the analytical model suggests that the SNR depends not only on the geometrical correction factor and input resistance but also on the bias current. It means that the *L*/*W* ratio has a dominated influence on the SNR at the fixed current bias. The model calculation indicates the optimal SNR can be achieved in the *L*/*W* range of 0.4–0.8, which is confirmed by the 3D TCAD simulation. The optimum *L*/*W* point in the range of 0.4–0.5 is suggested for the symmetrical cross-like Hall plate, which allows for layout optimization to attain the optimum device performance in the practical applications. For example in standard 0.18-μm CMOS process, the current-mode cross-like Hall plates can achieve the maximum sensitivity of about 6.7%/T when the *L*/*W* ratio is designed to 0.4–0.5. Meanwhile, in the crucial point of *L*/*W* = 0.5, the optimal SNR can also be achieved where the current-spinning technique is used to remove the device offset and 1/f noise. Therefore, the geometry optimization model proposed in this paper can provide us guidance for the optimal geometry design of the current-mode cross-like Hall sensors.

## 5. Conclusions

The influence of device geometry on the current sensitivity and SNR of the cross-like Hall plate working in the current mode has been investigated. Based on the conformal mapping calculation, a new analytical geometry optimization model is presented to analyze the current sensitivity and SNR dependence of *L*/*W* ratio and further to predict the optimal device geometry of the current-mode Hall plate. To verify the accuracy of the analytical model, 3D TCAD simulation was performed to study the effect of different *L*/*W* ratio on the current sensitivity and SNR. Theoretical model calculation agrees well with TCAD simulation results. It is revealed that an *L*/*W* ratio around 0.4–0.5 is beneficial for the optimal sensitivity and SNR for a current-mode cross-like Hall plate. The results show that in the 0.18-μm CMOS process, the cross-like Hall plate can obtain the maximum current-mode sensitivity of about 6.7%/T in the *L*/*W* ratio range of 0.4–0.5. In the same *L*/*W* ratio range, the current-mode Hall plate can also get the optimal SNR. Thus, the proposed analytical geometry optimization model provides an optimum geometry design rule for the current-mode cross-like Hall plates in the practical applications.

## Figures and Tables

**Figure 1 sensors-19-02490-f001:**
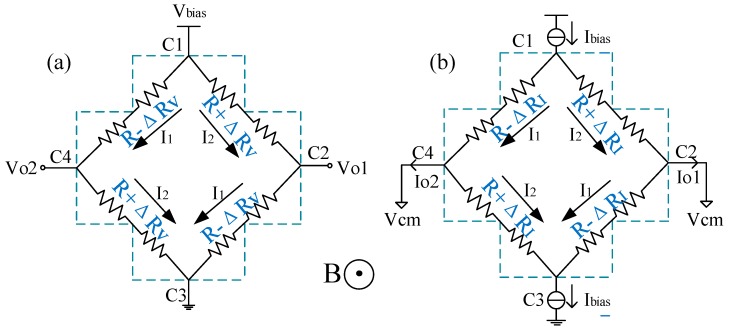
Wheatstone bridge model of the cross-like Hall plate. (**a**) voltage-mode; (**b**) current-mode.

**Figure 2 sensors-19-02490-f002:**
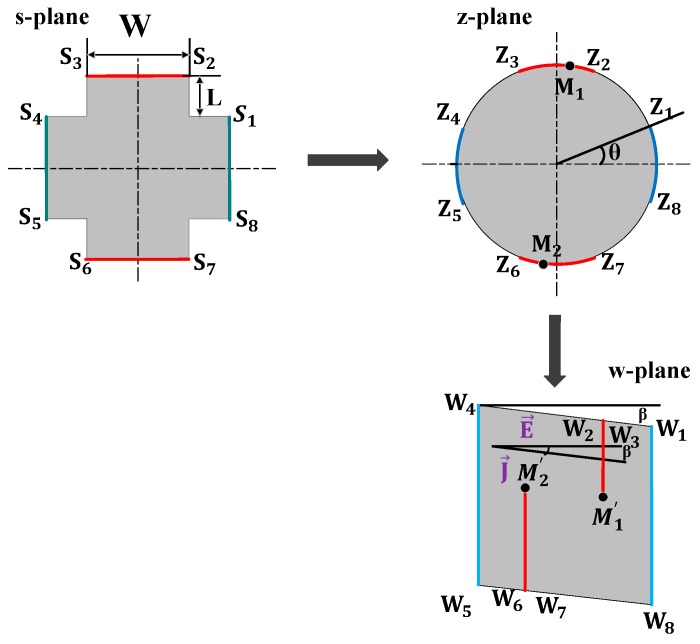
Schematic of the calculation procedure of *G*.

**Figure 3 sensors-19-02490-f003:**
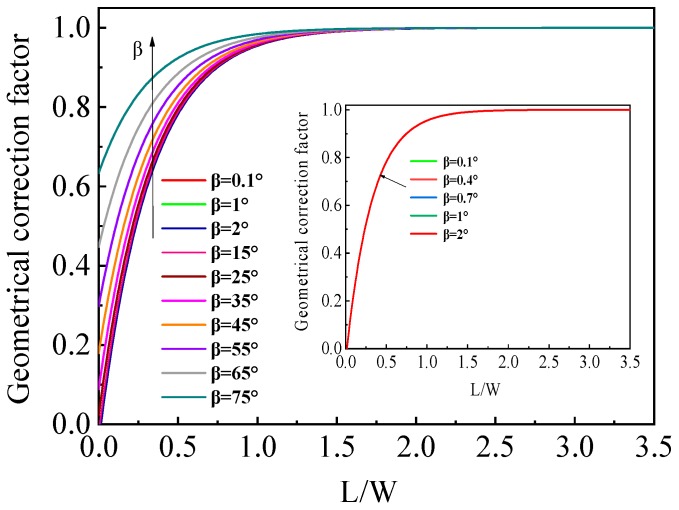
Geometrical correction factor *G* for the cross-like Hall plate versus *L*/*W* ratio at different Hall angles *β*.

**Figure 4 sensors-19-02490-f004:**
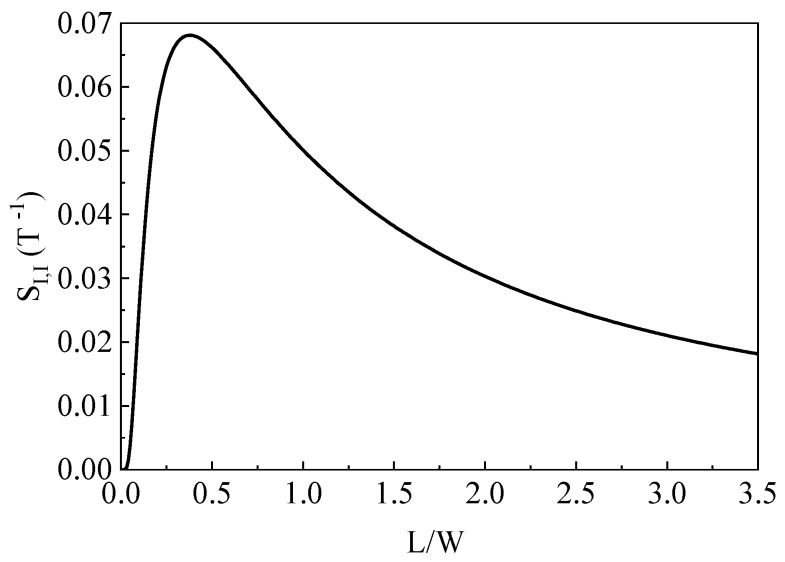
Current-mode sensitivity as a function with *L*/*W* ratio.

**Figure 5 sensors-19-02490-f005:**
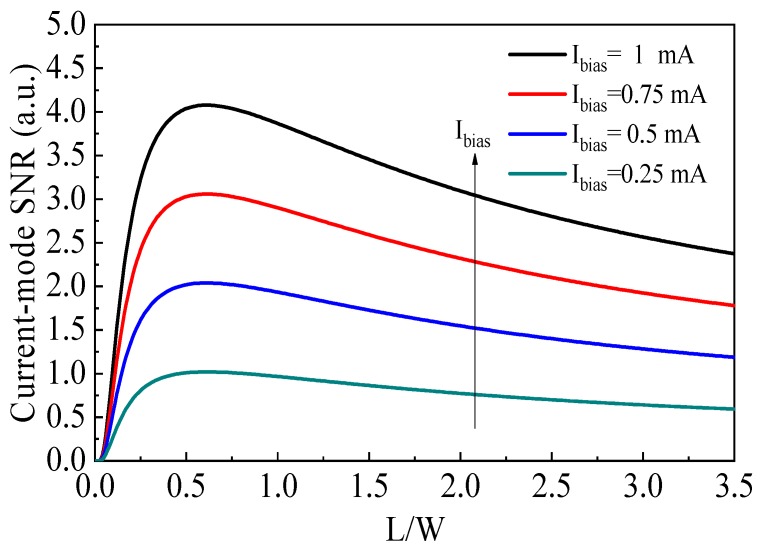
SNR as a function with *L*/*W* ration when a cross-like Hall plate is operating in the current mode.

**Figure 6 sensors-19-02490-f006:**
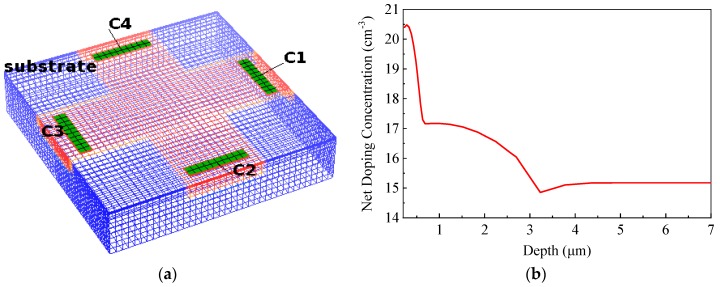
Technology Computer Aided Design (TCAD) simulation model using Silvaco Atlas tool. (**a**) 3D device simulation structure of the cross-like Hall plate; (**b**) 1D net doping profile in the device active region.

**Figure 7 sensors-19-02490-f007:**
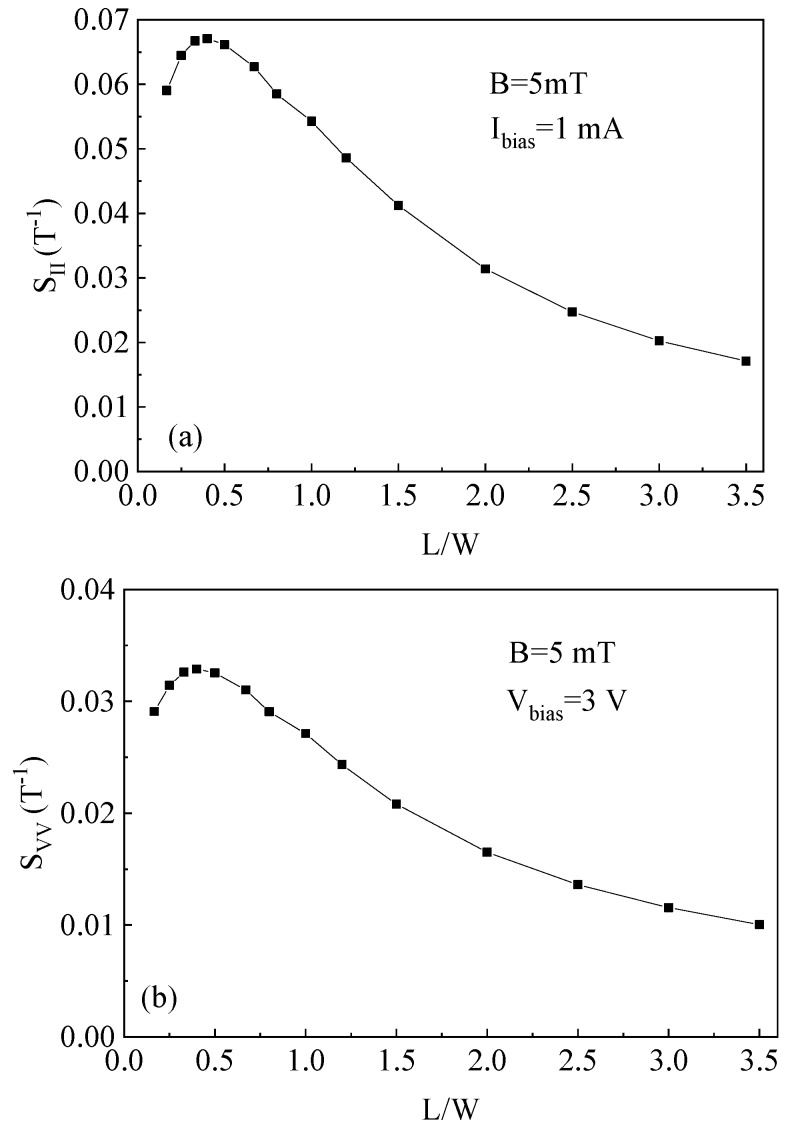
Simulated sensitivity versus different *L*/*W* ratios of the cross-like Hall plate. (**a**) current-mode, (**b**) voltage-mode.

**Figure 8 sensors-19-02490-f008:**
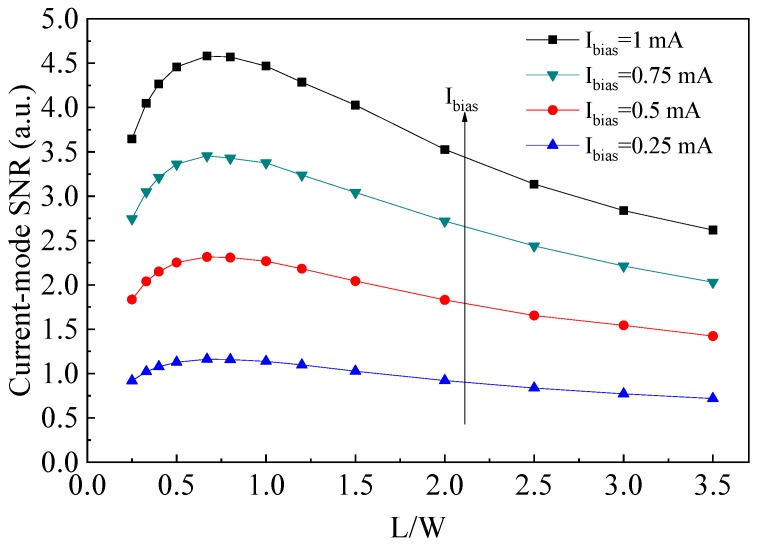
Simulated current-mode SNR varies with *L*/*W* ratio for different bias current condition under low magnetic field.

**Table 1 sensors-19-02490-t001:** Mapping parameters from the unit disk to rhomboidal region.

*k*	*α* _k_	*φ* _k_	*k*	*α* _k_	*φ* _k_
1	1/2 + β/π	θ	5	1/2 + β/π	π + θ
2	1/2 − β/π	π/2 − θ	6	1/2 − β/π	3π/2 − θ
3	1/2 + β/π	π/2 + θ	7	1/2 + β/π	3π/2 + θ
4	1/2 − β/π	π − θ	8	1/2 − β/π	2π − θ

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
