# Peer review of "An Analytical Geometry Optimization Model for Current-Mode Cross-Like Hall Plates"

_sensors, 2019, doi:10.3390/s19112490_

Round 1
Reviewer 1 Report
The authors present Hall sensor plates mathematical modeling based on analytical geometry and use it to optimize the sensor in terms of sensitivity and SNR, providing also a comparison with TCAD models. The manuscript is well written, with few spelling and grammar mistakes, indicated in the scanned document. The method is clearly defined, and the results are consistent with the hypothesis made.
I have some specific doubts and considerations:
- in equations (4) and (8), there is a variable B (magnetic flux density, I suppose) that had not been previously defined, and that actually does not seem to make sense. In fact, it would render the equations dimensionally inconsistent (as delta_R/R is dimensionless). Please clarify and/or correct if necessary.
- In line 150, after equation (20), the authors state that the adjusted curve has an accuracy better than 0.1%. However, as per the International Metrology Vocabulary, "accuracy" is a qualitative concept, and does not have a numerical value. The authors should use an adequate term to define the value of 0.1% error between the model and equation (20).
- throughout the manuscript, the units shall be separated by a space from the values (e.g. the correct is 1 T instead of 1T)

Author Response
Point 1: In equations (4) and (8), there is a variable B (magnetic flux density, I suppose) that had not been previously defined, and that actually does not seem to make sense. In fact, it would render the equations dimensionally inconsistent (as delta_R/R is dimensionless). Please clarify and/or correct if necessary.
Response 1: Thank you for your comment. As you said, the variable B denotes the magnetic flux density. In the revised manuscript, the variable B is defined. In addition, the dimension of sensitivity in Equations (4) and (8) should be 1/T. The Equations (4) and (8) have been corrected.
Point 2: In line 150, after equation (20), the authors state that the adjusted curve has an accuracy better than 0.1%. However, as per the International Metrology Vocabulary, "accuracy" is a qualitative concept, and does not have a numerical value. The authors should use an adequate term to define the value of 0.1% error between the model and equation (20).
Response 2: Thank you for your suggestion. The term “accuracy” is only a qualitative concept. In the revised manuscript, this term has been modified as follows: the relative deviation between the G model and the fitted formula (20) is less than 0.1%.
Point 3: throughout the manuscript, the units shall be separated by a space from the values (e.g. the correct is 1 T instead of 1T)
Response 3: Thank you for your suggestion. These format problems have been corrected in the revised manuscript.
Reviewer 2 Report
Paper can be accepted after the following corrections:
1. The implementation of the method of simulation should be clearly stated.
2. Authors should consider the comparison of the results with the results of finite Elements Modelling (FEM) previously presented in the literature for cross-shaped Hall sensors.
3. Equations 9, 10, 18, line 181, etc. should be re-edited. Please clearly present R_square parameter.
4. Conclusions should be clearly stated considering practical applications. Please clearly indicate the quantitative results of analyses.
Author Response
Point 1: The implementation of the method of simulation should be clearly stated.
Response 1: Thank you for your suggestion. In the revised manuscript, the implementation of the TCAD simulation method has been clearly stated as follows:
First of all, based on a standard 0.18 μm process a 3D device simulation structure was established according to the process and geometric parameters such as the distribution of N-well doping concentration, N-well depth, contact size, and N-well size, etc. Secondly, the device DC simulation was carried out where the two bias currents of the same size are applied to the input terminals, and the output Hall signal and noise are measured between the sensing terminals. Finally, the current sensitivity and signal-to-noise ratio (SNR) was calculated according to the simulation data.
Point 2: Authors should consider the comparison of the results with the results of finite Elements Modelling (FEM) previously presented in the literature for cross-shaped Hall sensors.
Response 2: Thank you for your suggestion. We make comparison of the TCAD simulation results with the results of finite Elements Modelling (FEM) previously presented in the literature [4]. It is found that the obtained magnetic sensitivity from TCAD simulation shows higher accuracy. It is due to the fact that the key model parameters used in TCAD simulation, such as N-well conductivity, electrons concentration, Hall coefficient, can be accurately calculated by the simulator according to the distribution of impurity concentration and device material characteristics. However, in the FEM simulation, these key parameters need to be defined in advance, so it is difficult to consider the influence of actual process. In the revised manuscript, we make the comparison of the TCAD results with the FEM simulation results reported in the literature for cross-shaped Hall sensors
Point 3: Equations 9, 10, 18, line 181, etc. should be re-edited. Please clearly present R_square parameter.
Response 3: Thank you for your suggestion. In the revised manuscript, Equations 9, 10, 18, line 181, etc. have been re-edited and the R_square parameter is clearly defined.
Point 4: Conclusions should be clearly stated considering practical applications. Please clearly indicate the quantitative results of analyses.
Response 4: Thank you for your suggestion. In the revised manuscript, the quantitative result analysis is made and the practical application of the proposed model is also clearly stated as follows:
The optimum L/W point in the range of 0.4-0.5 is suggested for the symmetrical cross-like Hall plate, which allows for layout optimization to attain the optimum device performance in the practical applications. For example in standard 0.18 μm CMOS process, the current-mode cross-like Hall plates can achieve the maximum sensitivity of 6.7%/T when the L/W ratio is designed to 0.4-0.5. Meanwhile, in the crucial point of L/W=0.5, the optimal SNR can also be achieved where the current-spinning technique is used to remove the device offset and 1/f noise. Therefore, the geometry optimization model proposed in this paper can provide us guidance for the optimal geometry design of the current-mode cross-like Hall sensors.
Round 2
Reviewer 2 Report
Paper was corrected accordingly to the review and can be accepted in the present state.